# InterTwin: Deep Learning Approaches for Computing Measures of Effectiveness for Traffic Intersections

Yashaswi Karnati , Rahul Sengupta  and Sanjay Ranka *

Department of Computer and Information Science and Engineering, University of Florida, Gainesville, FL 32608, USA; yashaswikarnati@ufl.edu (Y.K.); rahulseng@ufl.edu (R.S.)
* Correspondence: yash.karnati.io@gmail.com

**Abstract:** Microscopic simulation-based approaches are extensively used for determining good signal timing plans on traffic intersections. Measures of Effectiveness (MOEs) such as wait time, throughput, fuel consumption, emission, and delays can be derived for variable signal timing parameters, traffic flow patterns, etc. However, these techniques are computationally intensive, especially when the number of signal timing scenarios to be simulated are large. In this paper, we propose InterTwin, a Deep Neural Network architecture based on Spatial Graph Convolution and Encoder-Decoder Recurrent networks that can predict the MOEs efficiently and accurately for a wide variety of signal timing and traffic patterns. Our methods can generate probability distributions of MOEs and are not limited to mean and standard deviation. Additionally, GPU implementations using InterTwin can derive MOEs, at least four to five orders of magnitude faster than microscopic simulations on a conventional 32 core CPU machine.

**Keywords:** deep learning; deep simulators; traffic signal timing optimization; intelligent transportation systems





## 1. Introduction

Urban traffic control [1] is one of the most important and challenging issues facing cities and requires practically effective and efficient solutions. The increasing volume of traffic in cities has a significant effect on road traffic congestion and consequently the travel time of road users. Optimization methods have been successfully used to reduce travel time by optimizing the signal timing parameters. These methods try to derive the signal timing parameters such as green splits, cycle lengths offsets, etc. based on the traffic patterns for a given duration for a given intersection. A number of scenarios—each corresponding to a combination of different signal timing parameter values—is generated and simulation is performed to generate Measures of Effectiveness (MOEs) such as throughput and delays. MOEs are derived using macroscopic or microscopic simulation methods, the latter being more computationally intensive but generally more accurate. When these MOEs have to be computed for a large number of scenarios, the process can become computationally intensive. This is further accentuated when optimizing for a corridor or a network as the number of combinations increase exponentially. Methods such as ReTime address this by parallelizing VISSIM [2] instances on multiple processors.

Our objective in this paper is to develop a deep neural network approach that can compute MOE distributions that are generated by microscopic simulators. Microscopic simulators generate trajectories of each vehicle (effectively location at every time step) along with car-following and lane-changing models. This information can then be used to compute MOEs, such as throughput, energy requirements, as well as emission and delays based on different signal timing scenarios. Examples of simulators that can be used for this purpose include VISSIM [2], SUMO [3], and AIMSUN [4]. Our approach does not require generation of trajectories and can directly compute MOE distributions. Once a distribution is available, statistics such as mean, variance, 90th percentile, etc. of

an MOE can be easily calculated. We demonstrate the feasibility of achieving this for a single intersection with high accuracy. The main advantage of our approach is that when implemented on modern GPU-based processors, it is four to five orders of magnitude faster than running the microscopic simulators on a typical 32-core CPU machine. Clearly, if the actual vehicle trajectories are required on a given intersection, our method is not useful or appropriate.

Our deep learning approach (called InterTwin as a short form for Intersection Twin) uses a large amount of data from different intersection topologies and signal timing plans, so as to capture the underlying traffic behavior at an intersection. While several real-world datasets exist [5], they are limited to only currently-used signal timing plans. We use microscopic simulators in conjunction with real world data to derive MOE distributions for a variety of scenarios that encompass a broad range of signal timing plans. Using this data, we built our models. The following are the key contributions of our work:

1. We develop a novel two-module deep learning approach that captures the intrinsic properties of traffic behavior at an intersection. The first module corresponds to a spatial graph convolution that is used to extract spatial features from the detector waveforms leveraging the relationship between intersection lanes and signal timing phases. This makes our modeling relatively independent of the intersection topology. The second module is an encoder-decoder with temporal attention architecture, to capture the temporal dynamic behavior of the traffic flow for each phase based on the signal timing plan. These two modules are stacked together for obtaining the final prediction.

2. We show that the InterTwin-trained models are able to accurately predict MOE distributions generated by traffic simulators. After training, when these models are used in inference mode, these models are four to five orders of magnitude faster compared to microscopic simulations. Additionally, it can model multiple intersection topologies without painstakingly redrawing a new base map for each intersection (that is typically required by a microscopic simulator).

3. For training our models, we use data generated using a significant extension of SUMO [3], an open source microscopic traffic simulator to make the data generation more realistic. We use real-world recorded data from high resolution loop detectors for input traffic patterns. Additionally, we have developed a new module that uses ring and barrier implementation along with arrival and departure information at the advanced and/or stop bar loop detectors along with signal timing information using techniques described in [6] and use them in our simulation to generate high fidelity MOEs that are reflective of data collected from real intersections. Additionally, we suitably vary signal timing parameters for these patterns to generate potentially viable counterfactuals. This results in our methods being able to generalize beyond what is typically used in actual practice and ensures that the models trained can predict robustly for a wide range of signal timing parameters. We also simulate a variety of intersection basemaps and behaviors, and estimate different measures of effectiveness, such as queue lengths, travel times, and wait times.

We present extensive experimental results to demonstrate the effectiveness of our approach. This includes comparison with other traffic-related deep learning models (that were not necessarily developed for predicting MOEs). We also show our model can be effectively used finding trade-offs between multiple MOEs at an intersection while evaluating different signal timing plans. Although the focus of this paper is on a single intersection, we believe that the approach can be extended to corridors and networks and will be part of our future work.

The rest of the paper is outlined as follows. Section 4 describes the proposed framework and architecture of the deep learning model presented. The ring and barrier implementation and a parallel framework for Simulator of Urban Mobility (SUMO) is described in Section 3. The data generation mechanism is also detailed in this section. Experimental

results are provided in Section 5, and a case study that uses this framework on a real world intersection is presented in Section 6, with conclusions presented in Section 7.

## 2. Related Work

The idea of using deep neural networks to emulate physics-based simulations is not completely new. A deep learning framework with graph neural networks is proposed in [7] to simulate complex physical systems involving fluids, rigid solids, and other deformabale objects. This idea of building neural network-based emulators for physics-based simulations in different domains—high energy physics, climate science, astrophysics, and seismology has been explored in [8]. The authors propose an algorithm based on neural architecture search to approximate/emulate the simulations using deep neural networks that are accurate and orders of magnitude faster (up to 2 billion times). The main advantage of neural network emulators is that they are orders of magnitude faster and can be useful for extensive parameter exploration and very large scale analysis. Based on our detailed literature survey, we believe this is this is the first work proposing neural network emulators for traffic microscopic simulations to compute measures of effectiveness.

Machine learning techniques, including deep neural networks have been successfully applied to traffic state data in the past literature. Existing literature on predicting traffic state include either predicting either volumes or performance measures (such as travel times, wait times, queue length, etc.). The prediction horizon could be either at a sub-cycle level or at aggregate intervals (5 min to 1 h). We outline this work below.

A hybrid method incorporating filtering-based empirical mode decomposition is proposed in [9]. A deep learning-based method with non-parametric regression is proposed in [10], a novel deep neural network architecture with multisegents (recurrent and convolutional layers) is proposed in [11]. A deep generative model based on generative adversarial networks is proposed in [12]. A hybrid method based on linear programming, fuzzy logic, and multi layer perceptron is proposed in [13]. Some of the recent deep learning architectures proposed for spatio-temporal forecasting of traffic state data include temporal graph convolution network [14], spatio temporal residual graph attention network [15], and spatio temporal residual graph attention network [15]. Some relevancy to our work is proposed in [16,17], which uses real world data to drive the microscopic simulations to compute performance indices such as travel time, emission performance, etc. that are used for the network. The main difference of our work from the existing work is two fold:

- We propose deep neural networks for estimating the distribution of performance measures instead of doing the microscopic simulations. Our methods are at least four to five orders of magnitude faster;
- Our models can also predict the performance measures for counterfactual signal timing plans, i.e., for a given input traffic and also for different cycle times and green splits (signal timing parameters). This can be useful to study the impact of different signal timing parameters.

The key observation is that neural network models can be used for computationally efficient parameter exploration. One important application in case of traffic intersections is that this approach can be used to find signal timing parameters for each of the intersection in a city corridor/network that satisfies a particular objective.

## 3. Simulator for Dataset Generation

MOE prediction algorithms must be trained using a large amount of realistic datasets for a wide variety of parameters for them to be predictive in realistic scenarios. These parameters include intersection topology, signal timing parameters, and input traffic distributions for all directions. The modern road network infrastructure (signal controllers and detectors) continuously generates data that opens up a new space of possibilities for using them for deriving realistic signal timing parameters, input traffic patterns, and a combination thereof. In particular, high resolution signalized intersection controller logs comprise of a listing of timestamped events at 10Hz using induction loop detectors [18]



at stop-bar and in some cases at upstream (40 m–100 m) locations. The data collected by such systems [19] at each intersection can be broadly divided into signal timing data and loop detector data. The former consists of traffic movement timing for different phases (including pedestrian calls), while the latter consists of arrival, departure, and occupancy information of loop detectors. Table 1 shows a sample of high resolution raw data collected by the loop detectors. It also comes with metadata that describe different event codes and event parameters; for example, eventcode 81 indicates a vehicle departure, and event code 2 indicates start of green phase. An event parameter identifies the particular detector channel or phase in which the event was captured. This data can be used to drive simulations and capture performance measures such as travel times, wait times, queue lengths, etc. One of the novelty of our approach is that we use this real-world high-resolution data to drive our simulations, the focus being to be able to run simulations for different signal timing parameter combinations.

**Table 1.** Table showing raw event logs from signal controllers. Most modern controllers generate these data at a frequency of 10 Hz.

| Signal ID | Times Tamp | Event Code | Event Param |
|---|---|---|---|
| 1490 | 2018-08-01 00:00:00.000100 | 82 | 3 |
| 1490 | 2018-08-01 00:00:00.000300 | 82 | 8 |
| 1490 | 2018-08-01 00:00:00.000300 | 0 | 2 |
| 1490 | 2018-08-01 00:00:00.000300 | 0 | 6 |
| 1490 | 2018-08-01 00:00:00.000300 | 46 | 1 |
| 1490 | 2018-08-01 00:00:00.000300 | 46 | 2 |
| 1490 | 2018-08-01 00:00:00.000300 | 46 | 3 |

Vehicle actuations at these detectors can be considered as pulse waveform, can represent the arrival/departure pattern at a given detector. For a given traffic stream on an approach (based on traffic from a neighboring interaction), the waveform at the advanced and stop bar detector, in general, is dependent on the signal phase timing for that intersection. This makes it challenging to develop combinations where we would like to predict MOEs for combinations that are not available in the datasets. To address this problem, we leverage our earlier work [6] that uses neural network architectures for inflow waveform reconstruction using stopbar, advance detector, and signal timing data. The inflow waveform can be thought of as the incoming waveform that has exited the upstream intersection and is still sufficiently far away from the intersection of interest and thus is not yet affected by the queues and signal timing plan of the intersection of interest. The simulations can be run using this inflow waveform with different candidate signal timing plans.

We use Simulator of Urban Mobility (SUMO) [3], an open source microscopic traffic simulator as our choice of physical simulator. The overall workflow for dataset generation is as follows:

- Reconstruct the unperturbed inflow waveform using stop bar, advance detector actuations, and signal timing information;
- Extract signal timing details from controller logs;
- Create the intersection base map in SUMO based on Google Maps satellite imagery using SUMO NETEDIT;
- The signal timing plan is perturbed within feasible limits to generate viable counterfactual simulations;
- Run multiple simulations in parallel for different signal timing plans;
- The vehicle traces along with detector output files are parsed to store route wise distribution of travel times, wait times along with waveforms at stopbar, advance detectors, and signal timing information.

The signal timing control type is either pre-timed or actuated in SUMO, but for effective operation, ring and barrier control is often employed in practice (Figure 1). To

address this, we have developed a dual-ring and barrier controller module for SUMO to mimic real world intersection controller.

It is convenient to think of the time period of an intersection in terms of cycles. A complete cycle consists of a predetermined sequence of traffic phases, where a traffic phase consists of green time allocated to a set of lanes simultaneously for non-conflicting movement of traffic through an intersection. The cycle length is the time interval of a complete cycle and can vary from one cycle to the next. Within a cycle, a green split is the time interval allocated for a particular traffic phase. A ring and barrier controller allows us to separate the 8 lane-movements (i.e., phases 1–8) into two concurrency groups, with one group with the major street movements and one for minor street movements (or more generally, opposing direction flows).

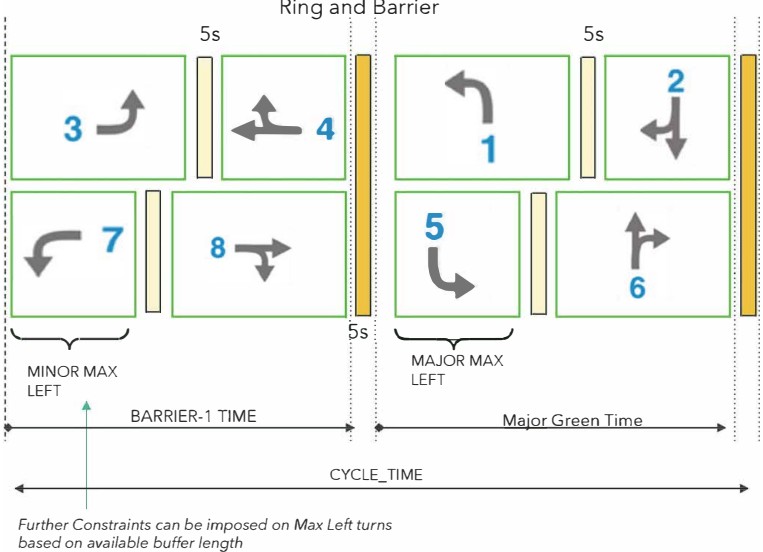

**Figure 1.** Ring and barrier controller allows us to separate the 8 lane-movements (i.e., phases 1–8) into two concurrency groups, with one group with the major street movements and one for minor street. Key signal timing parameters include cycle time, barrier time, max green splits, etc. For a given input traffic flow, the signal timing parameters are varied to generate different scenarios.

This approach allows us to generate MOEs for a number of combination of traffic patterns for each approach and signal timing parameters (CYCLE_TIME, Barrier-1 times, etc., Figure 1). For this work, we have used 1 million exemplars for training our neural network model. The simulation environment makes use of multi-threading where up to several instances of SUMO will be running in parallel, each simulating separate combinations of input traffic patterns and signal timing plans.

## 4. Proposed Framework

Our focus is to train deep neural networks to be able to estimate performance measures such as wait times, travel times, etc., for a given input traffic and also for different cycle times and green splits (signal timing parameters). These trained models should be able to approximate the dynamics of physical simulators, when used in inference mode has the following advantages:

- The models are four to five orders of magnitude faster compared to simulations;
- The model can be used for multiple intersection topologies;
- These models can also be used to bootstrap the training of reinforcement learning-based optimization algorithms.

Effectively, the model captures the interrelationships between the various traffic, signal timing, and topology parameters and their impact on MOEs and allows for understanding the impact of a variety of signal timing parameters on MOEs for a given set of input traffic

patterns for a given intersection topology. Additionally, our model generates a distribution rather than summary statistics. The overall workflow for training the neural network is shown in Figure 2. The data corresponding to different counterfactual signal timing plans is generated using microscopic simulators to train deep neural networks. For the rest of this section, for ease of description, we use distribution of wait times as an example of our performance measure. However, our framework is general enough to be used for other measures (or multiple measures) as well. The deep learning model takes input traffic as input and outputs distribution of wait times for different signal timing plans.

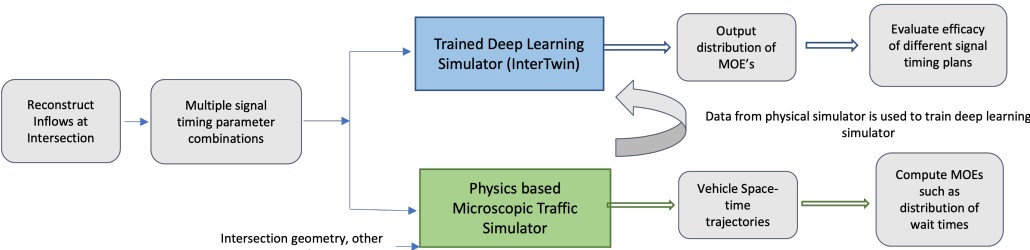

**Figure 2.** Overall workflow for training the neural network. The data corresponding to different counterfactual signal timing plans is generated using microscopic simulators to train deep neural networks. The trained neural networks can replace the simulators for predicting MOEs such as wait times and are four to five orders of magnitude faster. MOEs: Measures of Effectiveness.

The input traffic waveforms are represented as 1-D vectors, each with T components. Here T refers to the length of time a particular detector's data (arrival volumes) is being considered, with each component being aggregated at a 5-s level. In our work, $T = 72$, i.e., each data vector corresponds to 360 s of data. The output is the distribution of wait times for the 360-s window. The maximum wait time in our dataset is 2000 s, the wait times are binned into bins of size 10 s. So, the output is represented as a vector of size 200. Our proposed model, InterTwin, shown in Figure 3, has two main modules:

1. Spatial Graph Convolution (Spatial GC) is used to extract spatial features from the detector waveforms where the connectivity information of the intersection is incorporated;
2. The Encoder Decoder with Temporal Attention (EDTAM) module is used to capture the temporal dynamic behavior of traffic flow. These two modules (GC and EDTAM) are stacked together for obtaining the final prediction. The details of each module is as follows.

We now describe each of these modules in detail.

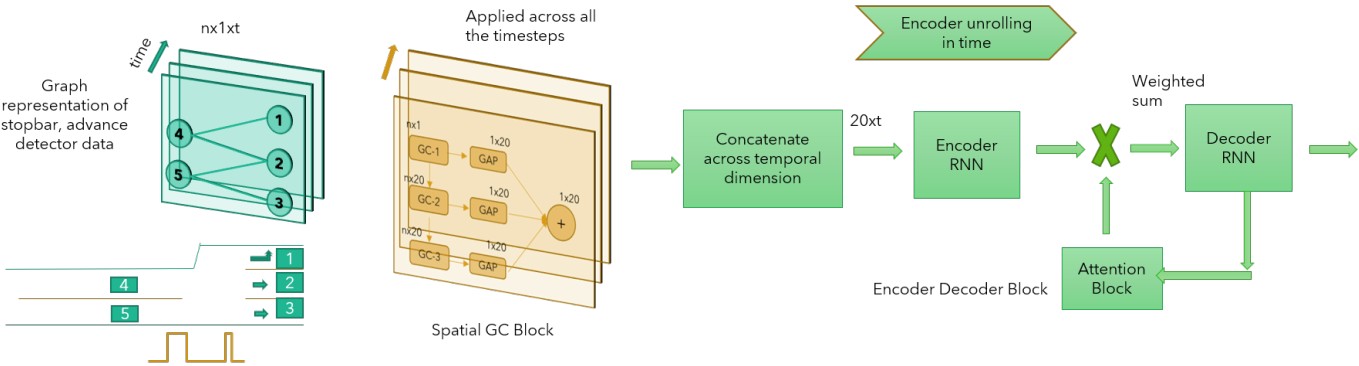

**Figure 3.** Architecture of the proposed InterTwin model. Spatial Graph Convolution (Spatial GC) is used to extract spatial features from the detector waveforms where the connectivity information of the intersection is incorporated. The encoder-decoder block is used to capture temporal dynamic behavior of the traffic flow.

### 4.1. Spatial Graph Convolution

The traffic movement on an intersection can be naturally represented in a graph structure, where each lane is represented as a node and an edge connects two nodes if one lane is feeding into another. We represent each detector (stop bar or advance) as a node and an edge exists between two nodes if there is a direct connection between them.

Graph convolutions provide a natural and meaningful way to extract features that can be used by higher layers where connectivity/spatial information has to be incorporated. Unlike standard Convolution Networks (CNNs) or Recurrent Networks (RNNs), graph convolutions can operate on irregularly structured data and can easily exploit the spatial structure of the intersection. Although CNNs are useful to generate adjacent spatial features, they are limited to regular/fixed grids).

In addition to being capable of handling different intersection topologies using graphs, representing an intersection as a graph structure can help the model learn from the spatial structure of the intersection, thereby providing relational inductive bias to the model. This also helps the model to generalize better for unseen intersections during the training phase [20]. Spatio-temporal graph convolution networks have been successfully used for city-scale traffic forecasting [21], where they show that their model outperforms other state-of-the-baseline models on a real world traffic dataset. Our approach models traffic at a much finer granularity as compare to the work in [21] (5 s versus 5 min).

Several methods have been proposed for generalizing convolutions on graphs [22], and they can be broadly classified into spectral-based or spatial-based methods. The spectral approach tend to capture the global structure of the graph more accurately than spatial methods. Unfortunately, spectral convolution methods require the eigen-decomposition of the graph laplacian, which is computationally expensive. There are several good approximation approaches available for spectral-based graph convolutions [23–25]. Considering the large size of our dataset, we employ the approach proposed by [23] based on first order approximation of localized spectral filters.

Consider an intersection is represented as graph $\mathbf{G} = \{\mathbf{V}, \mathbf{E}, \mathbf{A}\}$, consists of set of detectors $\mathbf{V}$ and $|\mathbf{V}| = n$, set of edges $\mathbf{E}$, and adjacency matrix $\mathbf{A}$. If there exists an edge between node $i$, node $j$ then $\mathbf{A}(i, j) = 1$, 0 otherwise. Each node of the graph is represented as a vector $\mathbf{x} \in \mathbf{R}^t$, this corresponds to arrival waveform with $t$ timesteps. Let $\mathbf{X} \in \mathbf{R}^{n \times t}$ be the node attribute matrix of the graph ($n$ detectors, $t$ timesteps). The notion of graph convolution operator in spectral graph convolution for a signal $X$ with kernel $\Theta$ can be seen as:

$$\Theta_{*\mathbf{G}} X = \Theta\left(\mathbf{U}\Lambda\mathbf{U}^T\right)X \tag{1}$$

where $\mathbf{U} \in \mathbf{R}^{n \times n}$ is the matrix of Eigenvectors of the normalized graph laplacian; $\Lambda \in \mathbf{R}^{n \times n}$ is the diagonal matrix of eigenvalues. As discussed earlier, we use approximate methods to compute $\Theta$. The convolution layer can be formulated as:

$$\mathbf{X}^{p+1} = \sigma\left(\tilde{\mathbf{D}}^{-\frac{1}{2}}\tilde{\mathbf{A}}\tilde{\mathbf{D}}^{-\frac{1}{2}}\mathbf{X}^p\Theta^p\right) \tag{2}$$

where, $\tilde{\mathbf{A}} = \mathbf{I} + \mathbf{A}$ is the adjacency matrix of graph $\mathbf{G}$, $\tilde{\mathbf{D}} = \sum_{j=1}^{n} \tilde{\mathbf{A}}_{ij}$ is the diagonal degree matrix of $\tilde{\mathbf{A}}$, $\Theta^p$ is trainable weight matrix of layer $p$, and $\sigma(.)$ is the activation function The above equation represents the layer wise propagation rule that makes up a single layer of the graph convolution. At a high level, this graph convolution operation aggregates the neighboring node features at each layer.

To summarize, we use this graph convolution layer to extract spatial features from the detector arrival waveforms. The proposed Spatial GC module consists of 3 stacked graph convolution plus Global Additive Pooling (GAP) layers. The feature map after each GAP layer is aggregated to obtain the final output of Spatial GC block (as shown in Figure 3). The weights of Spatial GC block are shared across all the timesteps, the output for each time step is concatenated across temporal dimension which in turn is fed into the EDTAM module.

### 4.2. Encoder Decoder with Temporal Attention

The encoder decoder model is a variant of Recurrent Neural Network (RNN) that has well suited modeling temporal sequences. These networks process input sequences within the context of their internal hidden state ("memory") in order to arrive at the output, the internal hidden state is an abstract representation of previously seen inputs. Thus, they are capable of modeling dynamic contextual behavior. We use Gated Recurrent Units (GRU) [26] as our choice of RNN in our implementation. The proposed EDTAM model consists of three building blocks—encoder, decoder, and temporal attention module. These are described below.

The encoder is an RNN that reads each timestep of the detector arrival waveform sequentially and updates its hidden state conditioned on the current input and its previous hidden state, Equation (3).

$$\mathbf{h_{e\langle t\rangle}} = f\left(\mathbf{h_{e\langle t-1\rangle}}, \mathbf{y_t}\right) \tag{3}$$

where $\mathbf{h_{e\langle t\rangle}}$, $\mathbf{y_t}$ are hidden state, input to the encoder at time step $t$ respectively. The hidden state of the encoder is stored after each time step, and the final output of the encoder is $\mathbf{H_{e\langle T\rangle}} = \left(\mathbf{h_{e\langle 1\rangle}}, \mathbf{h_{e\langle 2\rangle}} \ldots, \mathbf{h_{e\langle t\rangle}}\right)$.

The decoder is also an RNN that is trained to generate the output sequentially based on its input and hidden state. At each time step, the input to the decoder is conditioned on the decoder's output at the previous timestep, encoder's output, and current hidden state of the decoder. The Temporal Attention Module (TAM) acts as an interface between the encoder's outputs and the input of the decoder.

The Temporal Attention Module (TAM) is a fully-connected network with the inputs being the decoder's output at a previous time step and its hidden state. The output of TAM is a vector, attention scores, which is used to compute the weighted sum of the encoder's hidden states. This weighted sum is fed as input to the decoder, the attention helps the model to focus on specific parts of the encoder's outputs for prediction at each step.

The final hidden state of the encoder is used to initialize the hidden state of the decoder. The attention scores for predicting at timestep, $t$ are calculated as follows. Let $h_d\langle t\rangle$, $z_t$ represent the decoder's hidden state, output respectively for time step $t$

$$A_T = \frac{\exp\left(\mathbf{w}_j[h_{d\langle t\rangle} + z_{t-1}]\right)}{\sum_{j'=1}^{K} \exp\left(\mathbf{w}_{j'}[h_{d\langle t\rangle} + z_{t-1}]\right)} \tag{4}$$

where $w_j$ are the rows of learnable weight matrix $\mathbf{W}$, and $A_T$ represents the computed attention scores. The input to the decoder is the dot product of these attention scores with the hidden states of the encoder, $A_T.H_{e\langle T\rangle}$.

### 4.3. Overall Network

The Spatial GC block and encoder decoder module are stacked together and trained end to end with a chosen loss metric, Adam optimizer. Implementation, training, and evaluation of the model was done using the PyTorch [27] library.

## 5. Experimental Results

We now present the experimental results of our approach and compare the performance of the proposed architecture with some of the recent deep learning architectures proposed for spatio-temporal forecasting. In particular, we compare the performance of our model against the following architectures.

- FCN: Fully Connected Network;
- RNN-FCN: Recurrent Neural Network followed by a fully connected layer;
- T-GCN: Temporal graph convolution network for traffic prediction as proposed in [14];
- STGCN: Spatio Temporal Graph Convolution Network for traffic prediction, as proposed in [21];

- ST-RGAN: Spatio Temporal Residual Graph Attention Network as proposed in [15].

As mentioned earlier, the output of our model is the distribution of wait times, $f_{WT}(wt)$, for a given input traffic pattern, signal timing parameters. Since we are trying to predict the distributions, a softmax operation is applied to the model output to convert it into a probability distribution. All the models are trained with the Adam optimizer and Mean Square Error (MSE) as the loss function. The different terms used to describe input, output variables is shown in Figure 4. Each of the waveforms aggregate counts for five-second intervals (the level of aggregation was chosen so that the number of vehicles in each interval is very small—generally less than three, but also large enough to keep the size of the network to be small).

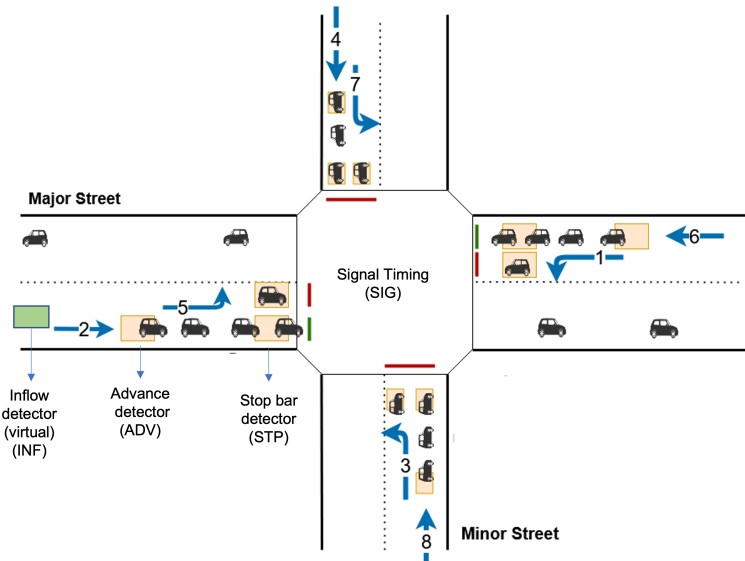

**Figure 4.** A typical intersection with 8 different directions of vehicular movement (phases 1 to 8). Vehicle waveforms are observed at Stop bar and Advance detectors (STP, ADV). STP, ADV, and INF corresponds to the traffic waveform at stopbar detector, advanced and inflow (500 m away from the intersection) aggregated at a 5-s interval. SIG corresponds to signal timing at a 5-s interval. STP, ADV, and SIG are typically available in ATSPM data for every intersection.

Table 2 shows that our methods are better for all the error measures: Root Mean Square Error (RMSE), Mean Absolute Error (MAE), and Mean Square Error (MSE). These results show that InterTwin is significantly better than FCN, STGCN, and T-GCN in terms of mean square error for MOE prediction. The key advantage of predicting distributions is that different summary statistics such as mean, median, and percentile values can be derived. Figure 5 shows actual vs. predicted scatter plot of the 50th, 70th percentile of wait time computed from the distribution. Figure 6 shows actual versus predicted distribution of wait time for a given inflow waveform, for different green time splits. These plots show that the models are able to capture the interrelationship between input traffic and signal timing parameters. We trained separate models each requiring different input traffic patterns for different Signal Timing waveform (SIG) using:

1. Only Inflow Traffic Waveforms (INF);
2. Only Advanced and Stopbar Waveforms (ADV and STP).

Table 3 shows the training and test errors for models with different input combinations. The accuracy is high when stopbar, advance waveforms are used as input compared to using only an inflow waveform. An important practical advantage of the model with only stopbar and advance waveforms as input is that both of these waveform are easily available in recorded controller logs at each intersection that support ATSPM. Thus, this model can be very effectively used to infer wait time distributions from recorded controller log data

in real world. Unfortunately, this model cannot be used directly for computing MOEs for different signal timing plans. The latter is generally required for optimization purposes.

For optimization, it is more practical to use the INF waveform and SIG waveforms as input as this model allows for varying signal timing plans. As discussed earlier, INF waveforms for each approach can be computed using neural networks that use ADV and STP waveforms along with SIG waveforms. For a given inflow, we can generate multiple candidate signal timing plans, use the trained model in inference mode to evaluate each timing plan based on the distribution of wait times. Using these trained models for MOEs estimation is highly scalable compared to simulation-based approaches.

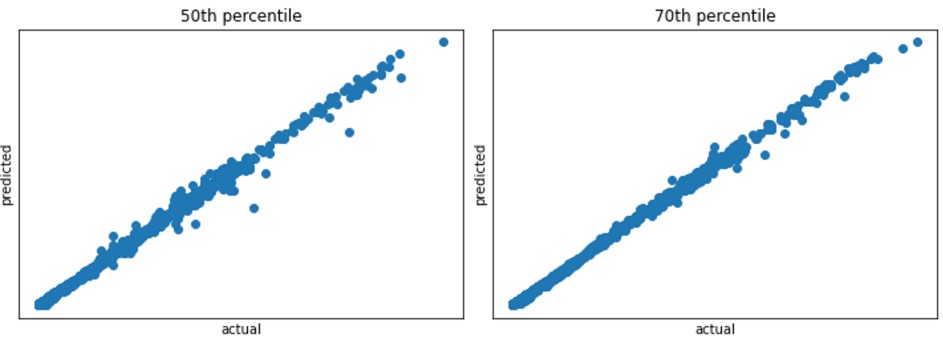

**Figure 5.** Summary statistics such as 50th/70th percentile etc. can be computed from the predicted distribution. Actual vs. predicted scatter plot of 50th, 70th percentile of wait time computed from the distribution. The key advantage is that predicting distribution enables us to compute any statistic of interest.

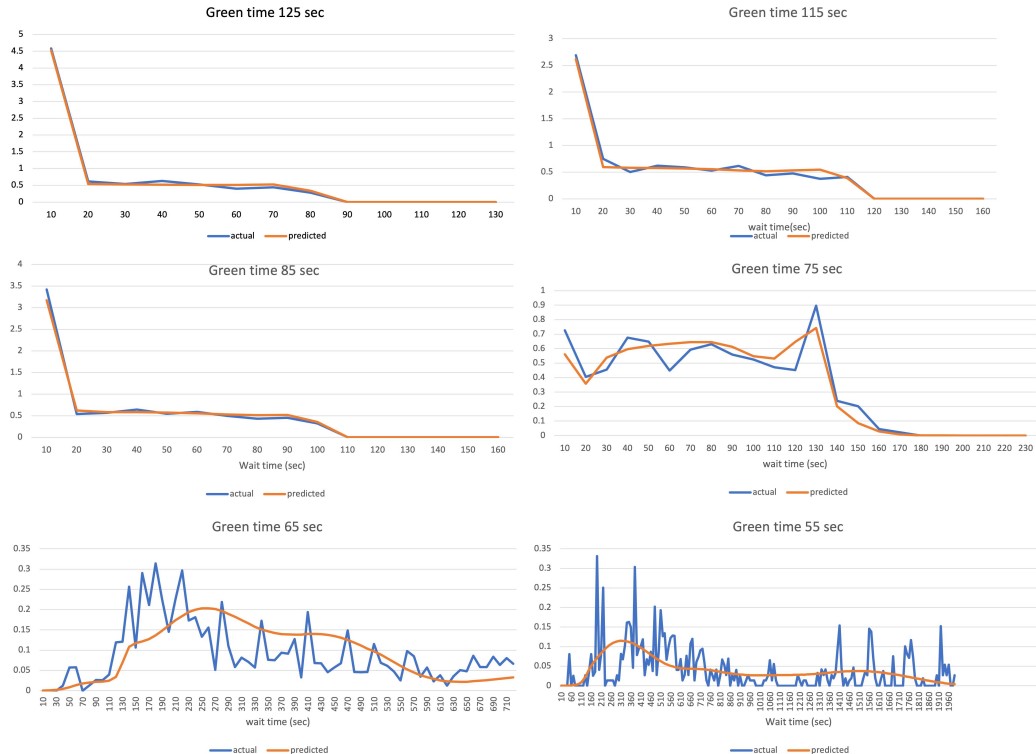

**Figure 6.** Actual vs. predicted distribution of wait time for different green time splits for a given input traffic. These plots show that the model is able to capture the interrelationship between input traffic and signal timing parameters.

**Table 2.** Table comparing performance of different models. The input to the models are ADV, STP, and SIG. This suggests that our InterTwin model has better accuracy compared to other model architectures. MSE: Mean Square Error, RMSE: Root Mean Square Error, and MAE: Mean Absolute Error.

| Model | MSE | RMSE | MAE |
|---|---|---|---|
| FCN | $1.5 \times 10^{-4}$ | 0.0084 | 0.003 |
| RNN-FCN | $2.2 \times 10^{-4}$ | 0.0091 | 0.0032 |
| T-GCN [14] | $1.2 \times 10^{-4}$ | 0.008 | 0.0026 |
| STGCN [21] | $1.5 \times 10^{-4}$ | 0.0085 | 0.0031 |
| ST-RGAN [15] | $5.4 \times 10^{-3}$ | 0.0165 | 0.0095 |
| InterTwin (ours) | $0.9 \times 10^{-4}$ | 0.0076 | 0.0023 |

**Table 3.** Comparison of model performance for different input parameters. This suggests that using STP, ADV has better accuracy compared to using INF waveform. The InterTwin model also has better accuracy. It is more practical to use INF along with SIG as INF waveforms are not affected by SIG and multiple signal timing parameters can be evaluated in parallel. Whereas, the other model (STP ADV SIG) can be useful to understand performance measures on recorded historical data.

| Model | Inputs | Train Error (MSE) | Test Error (MSE) |
|---|---|---|---|
| InterTwin | STP ADV SIG | $0.9 \times 10^{-4}$ | $0.9 \times 10^{-4}$ |
| InterTwin | INF SIG | $2.0 \times 10^{-4}$ | $2.1 \times 10^{-4}$ |
| FCN | STP ADV SIG | $1.3 \times 10^{-4}$ | $1.5 \times 10^{-4}$ |
| FCN | INF SIG | $3.0 \times 10^{-4}$ | $3 \times 10^{-4}$ |

For a given input traffic flow, to simulate 3200 different signal timing parameter combinations took more than 13,400 s on a 32 core machine. While the trained neural network model when used in inference mode generates output in 0.08 s for the same number of combinations (batch size 3200). This suggests that neural network models are at least four to five orders of magnitude faster.

## 6. Case Study

The trained models (that use INF and SIG waveforms) can be used to evaluate the efficacy of different signal timing plans for an intersection. This can then provide trade-offs for choosing a variety of signal timing parameters, including different green phase splits.

The 75th percentile of wait times as MOE is used for the rest of this discussion. In general, adding more time to the major street results in increased wait time on the minor street. The tradeoff can be clearly seen in Figure 7. It shows a scatter plot of 75th percentile of wait times for major vs. minor movements for different barrier-1 times (Figure 1). Based on this plot, a value of 60–70 s of green time may be appropriate as it minimizes the wait time on major street while not significantly impacting the wait time on the minor street. Of course a traffic engineer can look at these plots and other constraints to derive the optimal values. Given that our approach is computationally very inexpensive, optimal values can be derived separately for various hours of the day and day of the week combinations.

Table 4 shows a case study on a real intersection in Seminole County, Florida. We varied the barrier-1 time (keeping the cycle length fixed) to understand its impact on the wait time for traffic on major streets (NBT and SBT). These results show that changing the barrier time from 80 s to 60 s can lead to a 25% improvement in wait time for the major direction without significantly affecting the wait times on the minor street.

Rather than to provide one optimal signal timing plan, this framework can be extremely useful to practitioners to set green time splits for an intersection by understanding the trade offs for different hours of the day and days of the week.

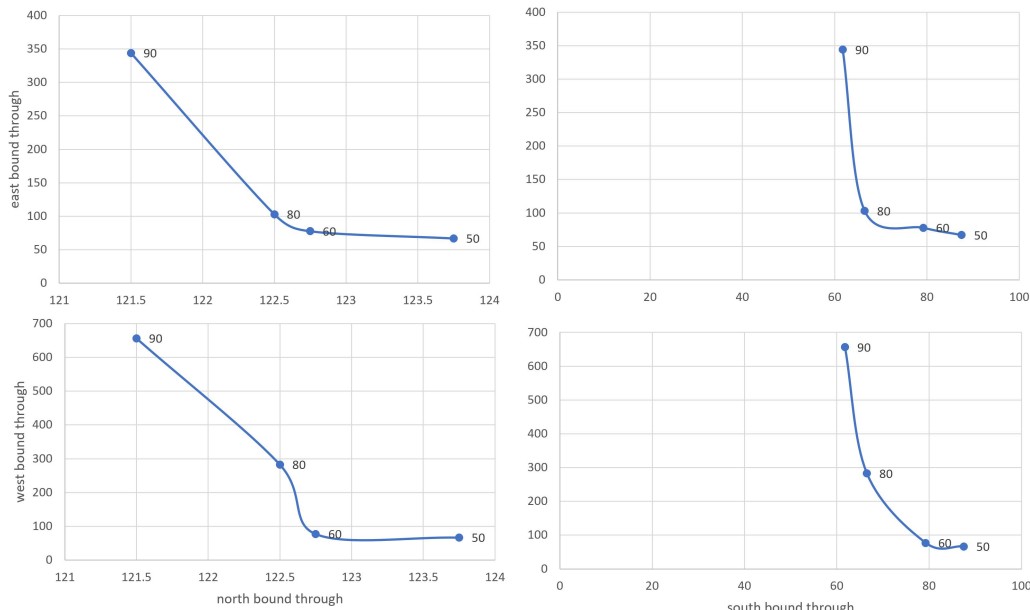

**Figure 7.** Scatter plot of 75th percentile of wait times for major vs. minor movements for different barrier-1 times. This can be useful to understand trade-offs in wait time for selecting different barrier-1 times for major vs. minor streets.

**Table 4.** Analysis on Intersection-1205 on 4 February 2019. Changing barrier-1 time from existing value 80 s to 60 s at 8:00 a.m. would improve the wait time on major direction by around 27%. NBT: North Bound Through, SBT: South Bound Through.

| Time | Barrier Time —New | Barrier Time —Old | % Improvement of Wait Time on NBT | % Improvement of Wait Time on SBT |
|---|---|---|---|---|
| 8:00 a.m. | 60 | 80 | 26 | 29 |
| 9:00 a.m. | 70 | 80 | 15 | 12 |
| 10:00 a.m. | 70 | 80 | 12 | 10 |
| 11:00 a.m. | 50 | 80 | 34 | 30 |
| 12:00 p.m. | 60 | 80 | 21 | 21 |
| 01:00 p.m. | 60 | 80 | 22 | 23 |
| 02:00 p.m. | 60 | 80 | 22 | 24 |
| 03:00 p.m. | 50 | 80 | 37 | 34 |
| 04:00 p.m. | 60 | 80 | 25 | 33 |
| 05:00 p.m. | 50 | 80 | 34 | 35 |

## 7. Conclusions

In this paper, we proposed InterTwin, a deep neural network architecture based on spatial graph convolution and encoder decoder recurrent networks that can predict the MOEs quickly and precisely. Rather than just predicting one or two statistics for a MOE (e.g., mean and standard deviation), our network can compute the entire distribution. Additionally, our models are four orders of magnitude faster than conducting detailed simulations.

Broadly, we presented two models based on the input waveforms that are used along with signal timing. The first one uses stopbar and advance waveforms. An important practical advantage of the model is that both of these waveforms are easily available in recorded controller logs at each intersection that support an ATSPM-based system. Thus, this model can be very effectively used to infer wait time distributions from recorded real-world controller log data. This model, however, cannot be used directly for computing MOEs for different signal timing plans.

The second model uses only inflow waveforms as input as this model allows for varying signal timing plans. For a given inflow waveform, we can generate multiple

candidate signal timing plans, use the trained model in inference mode to evaluate each timing plan based on the distribution of wait times.

We believe that this computationally-efficient approach can be extended to corridor optimization where the number of parameters is proportional to the product of parameters for each intersection on the corridor. We are currently developing such methods.

**Author Contributions:** Conceptualization, Y.K., R.S. and S.R.; methodology, Y.K., R.S.; validation, Y.K.; formal analysis, Y.K.; resources, S.R.; writing—original draft preparation, Y.K., R.S. and S.R.; writing—review and editing, Y.K. and S.R.; visualization, Y.K.; supervision, S.R.; project administration, S.R.; funding acquisition, S.R. All authors have read and agreed to the published version of the manuscript.

**Funding:** The work was supported in part by NSF CNS 1922782. The opinions, findings, and conclusions expressed in this publication are those of the authors and not necessarily those of NSF.

**Conflicts of Interest:** The funders had no role in the design of the study; in the collection, analyses, or interpretation of data; in the writing of the manuscript, or in the decision to publish the results.

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
