# Peer review of "InterTwin: Deep Learning Approaches for Computing Measures of Effectiveness for Traffic Intersections"

_applsci, doi:10.3390/app112411637_

Round 1

Reviewer 1 Report

The article highlights the lack of some key building blocks:
1- The literature review needs to be expanded to compare similar or competing studies. Studies to be compared should be presented in more detail and their deficiencies/disadvantages should be emphasized. This section will be helpful in highlighting the scientific contribution and explaining the difference of this paper.
2- Data collection and data structure should be presented in more detail. Only deep learning methods are used in the article. Why haven't machine learning methods that are easier to explain have been tried? Many disciplines have not given up on these algorithms. Is deep learning really necessary?
3- In the article, a benchmarking with competitor, similar or traditional methods and the results should be presented as a comparison table. Parameters claimed in the scientific contribution part should be recorded in this table and whether the targets/goals have been achieved or not should be presented. In addition, the codes and results of the study should be submitted with a link (at least to the reviewers and the editor).

Reviewer 2 Report

In this paper, it is found that the methods based on microscopic simulation is used to optimize the signal timing plans on traffic intersections are computationally intensive, especially when the simulation scene is complex. To solve this problem, the authors proposed a dual-ring and barrier controller module for SUMO to simulate the real world intersection controller, and proposed a deep neural network model based on spatial graph convolution and encoder-decoder recurrent networks to capture the inherent properties of traffic behavior. It can quickly calculate the probability distribution of MOEs, and compared with the microscopic simulations on CPU, the calculation is four to five orders of magnitude faster. The work of this paper will be highly practical and logical. In my view, this manuscript may be accepted after minor modification. And there are some problems to be further improved as well:

  1. There is at least one spelling error in the manuscript, such as, at line 70, the index of SUMO should be 3 instead of 7.
  2. In figure 6, it can be inferred that the interrelationship between input traffic and signal timing parameters can be captured by models when the time interval is longer than 75 seconds, but the fitting does not look so good when the time interval is 65 and 55 seconds. However, there is no analysis of this phenomenon in this paper. By the way, it might be better to add a legend to the diagram.
  3. I can not find the analysis of figure 5 in the experimental part.
